environmental science

catalytic activities, schwertmannite, few-layer graphene, Fenton-like reaction, sulfamethazine

**Author for correspondence:**
Dianzhan Wang
e-mail: dzwang@njau.edu.cn

This article has been edited by the Royal Society of Chemistry, including the commissioning, peer review process and editorial aspects up to the point of acceptance.

# Synthesis and assessment of schwertmannite/few-layer graphene composite for the degradation of sulfamethazine in heterogeneous Fenton-like reaction

Dianzhan Wang, Ye Gu, Zhaoshun Yang and Lixiang Zhou

Department of Environmental Engineering, College of Resources and Environmental Sciences, Nanjing Agricultural University, Nanjing 210095, People's Republic of China

DW, 0000-0002-3929-6924

Schwertmannite (sch), an iron oxyhydrosulfate mineral, can catalyse a Fenton-like reaction to degrade organic contaminants, but the reduction of Fe(III) to Fe(II) on the surface of schwertmannite is a limiting step for the Fenton-like process. In the present study, the sch/few-layer graphene (sch–FLG) composite was synthesized to promote the catalytic activity of sch in a Fenton-like reaction. It was found that sch can be successfully carried by FLG in sch–FLG composite, mainly via the chemical bond of Fe–O–C on the surface of sch–FLG. The sch–FLG exhibited a much higher catalytic activity than sch or FLG for the degradation of sulfamethazine (SMT) in the heterogeneous Fenton-like reaction, which resulted from the fact that the FLG can pass electrons efficiently. The degradation efficiency of SMT was around 100% under the reaction conditions of $H_2O_2$ 200–500 mg l$^{-1}$, sch–FLG dosage 1–2 g l$^{-1}$, temperature 28–38°C, and initial solution pH 1–9. During the repeated uses of sch–FLG in the Fenton-like reaction, it maintained a certain catalytic activity for the degradation of SMT and the mineral structure was not changed. In addition, SMT may be finally mineralized in the Fenton-like reaction catalysed by sch–FLG, and the possible degradation pathways were proposed. Therefore, the sch–FLG is an excellent catalyst for SMT degradation in a heterogeneous Fenton-like reaction.

# 1. Introduction

The heterogeneous Fenton-like process, one of the advanced oxidation processes, has been extensively used to remove organic contaminants from wastewater [1,2]. In a heterogeneous Fenton-like process, $H_2O_2$ is catalysed by solid catalysts to produce the hydroxyl radicals (–OH), which can effectively oxidize and decompose most organic contaminants [3,4]. Generally, the heterogeneous Fenton-like process has relatively wide availability and terrific catalytic properties [5,6]. To date, many kinds of solid catalysts, including $Fe^0$, $\alpha$-$Fe_2O_3$, Fe/UiO-66, Cu-ZSM-5, pyrite, etc. have been investigated to reveal their catalytic activities in a heterogeneous Fenton-like process for the removal of a broad range of contaminants [7–11].

Schwertmannite (sch) is a kind of Fe(III)-hydroxysulfate mineral formed in acid-mine drainage, acid-sulfate soils and sludge bioleaching environments and its formula can be expressed as $Fe_8O_8(OH)_{8-2x}(SO_4)_x$ ($x = 1$–1.75) [12,13]. Sch is rich in iron content, which makes sch a widely available heterogeneous Fenton-like catalyst for the treatment of wastewater. Wang *et al.* [14] used sch as a Fenton-like catalyst to degrade phenol and found that $100\ mg\ l^{-1}$ of phenol was degraded in 3 h. Meng *et al.* [15] reported that $1\ mg\ l^{-1}$ phenanthrene was completely removed from the solution in 3 h when using sch as a Fenton-like catalyst. Additionally, it has been already revealed that the Fenton-like process takes place on the surface of sch through the reaction between Fe(II) and $H_2O_2$ [6,14]. Given the fact that most iron on the surface of sch is Fe(III), Fe(II) should be generated by the reduction of Fe(III) during the Fenton-like process catalysed by sch [6,15]. However, the reduction of Fe(III) to Fe(II) on the surface of sch has a very low reaction rate, making it a limiting step for the degradation of organic contaminants in the Fenton-like process catalysed by sch [16,17]. Thus, it is reasonable to presume that increasing the reduction rate of Fe(III) to Fe(II) on the surface of sch may drastically promote the catalytic activity of sch in heterogeneous Fenton-like reactions.

Graphene is a kind of two-dimensional material with a flat single-layer of carbon atoms [18], which has large surface area and excellent electrical conductivity [19–21]. Many previous studies reported that graphene can be used as a catalyst carrier to enhance the performance of many catalysts, such as $Fe_3O_4$-GO, GO-$FePO_4$, GO-$Fe_2O_3$ and so on, in heterogeneous Fenton-like reactions [22–24], because the graphene can not only disperse the catalysts to prevent the catalyst agglomeration but also serve as electron donor–acceptor to enhance the conduction of electron, thus accelerating the oxidation and reduction reactions on the surface of catalysts [25,26]. Graphene-assisted materials have more stable and stronger electrical properties, even plant growth can be enhanced by graphene quantum dots [27–33]. Few-layer graphene (FLG) is constituted of 3–10 layers of single-layer graphene, which is also considered as a two-dimensional material with good physical and chemical properties, it can also be used in sensors [34,35]. However, most research on the graphene-supported-catalysts mainly focused on single-layer graphene. The performance of catalysts carried by FLG were seldom explored, even though the FLG was more convenient to produce [34].

Sulfamethazine (SMT), a sulfonamide antibiotic, has been widely used in veterinary practice owing to its broad antifungal spectrum [36,37]. It is noteworthy that most antibiotics used in animal feeding are discharged into farm wastewater, because of the very low absorption and use of antibiotics by livestock and poultry [38]. In addition, the antibiotics in farm wastewater cannot be effectively removed by the conventional biological wastewater treatment processes [39], and the rising concentrations of antibiotics in the environment may cause the spread of antibiotic-resistant bacteria and antibiotic-resistant genes that are seriously threating human beings' health [40]. Therefore, in the present study, SMT was selected as a target organic contaminant and the research objectives are (i) to synthesize sch/FLG composite (sch–FLG), (ii) to study the effects of reaction conditions including $H_2O_2$ dosage, catalyst dosage, initial solution pH and reaction temperature on the degradation of SMT during the reaction catalysed by sch–FLG, and (iii) to study the role of FLG in enhancing the catalytic activity of sch and the degradation mechanism of SMT during the Fenton-like reaction catalysed by sch–FLG.

# 2. Material and methods

## 2.1. Materials and reagents

$Fe_2SO_4 \cdot 7H_2O$, $H_2O_2$ solution (30%, v/v), and potassium iodide (KI) were purchased from Sinopharm Chemical Reagent Co., Ltd (China) at analytical grade. FLG was purchased from Suzhou Tanfeng Graphene Technology Co., Ltd (China). SMT (greater than or equal to 99%) and formic acid (high performance liquid chromatography (HPLC) grade) were purchased from Aladdin Company (China).

Methanol and acetonitrile were purchased from Merck Company (Germany) at HPLC grade. Deionized water was used throughout the present study.

## 2.2. Synthesis of schwertmannite/few-layer graphene composite

A weight of 22.24 g $Fe_2SO_4 \cdot 7H_2O$ was dissolved in 500 ml deionized water containing 0.5 g FLG, and then 6 ml $H_2O_2$ was dropwise added into the solution under stirring. The solution was then shaken for 24 h at 180 r.p.m. and 28°C in a rotary shaker. After that, the solution was filtered through a Whatman no. 4 filter paper to collect the precipitate. The precipitate was sequentially washed with acidified water (pH = 2.0) and deionized water for the respective three times, and then dried at 50°C until a constant weight. Meanwhile, the same procedures, except the addition of FLG, were carried out to chemically synthesize sch [41].

## 2.3. Characterization of catalysts

The morphology of sch–FLG was characterized by using high-resolution transmission electron microscopy (HRTEM, JEOL). The crystal structure of sch–FLG was characterized by using X-ray diffraction (XRD, Thermo Fisher XTRA) at a scanning rate of $10° \, min^{-1}$ in the $2\theta$ range of 10–70° with Cu-K$\alpha$ radiation ($\lambda = 1.5406$ Å) at room temperature. The surface elements of sch–FLG were characterized by using an X-ray photoelectron spectroscopy (XPS, Thermo Scientific ESCALAB 250Xi) system with Al K$\alpha$ radiation (Energy 1486.6 eV) and a laser Raman spectrometer (HR Evolution, HORIBA FRANCE SAS) in a spectrum scanning range of $100–4000 \, cm^{-1}$ using a solid-state semiconductor laser with $\lambda = 532$ nm. The Brunauer–Emmett–Teller specific surface area and Barret–Joyner–Halenda pore volume of sch–FLG was measured by using a $N_2$ adsorption–desorption method (Tristar 3000, Micromeritics). The chemical structure of sch–FLG was characterized by using Fourier transform infrared (FTIR, Thermo Nicolet 6700), and the samples were prepared with the powder pressing method in a potassium bromide pellet at room temperature.

## 2.4. Experimental procedures

The solution containing $5 \, mg \, l^{-1}$ of SMT was first prepared and the solution pH was adjusted to 3 using 1 M $H_2SO_4$. SMT degradation experiments were carried out in 35 ml glass vessels sealed with polythene film in a rotary shaker at 180 r.p.m. and 28°C. A $1 \, g \, l^{-1}$ of catalyst and 10 ml of SMT solution were added into each vessel, and then the degradation reaction was started up by adding $200 \, mg \, l^{-1} \, H_2O_2$ into the vessels. At the given reaction time intervals, the vessels were taken out correspondingly. After adding 30% (v/v) methanol to quench the reaction, the reaction solutions in vessels were filtered through a 0.22 µm filter film. After that, the solution was used to determine the concentrations of SMT, total iron, $Fe^{2+}$, $Fe^{3+}$, $H_2O_2$ and total organic carbon (TOC). To identify the intermediate products, the solution samples were pretreated using a solid-phase extraction method to concentrate the products. After the degradation experiments, the catalysts were collected, washed with deionized water, freeze dried and finally characterized by XPS and FTIR. In order to identify the presence of –OH, 10 mM of KI and 10% (v/v) of methanol were respectively added to scavenge –OH on the surface of the catalyst and –OH in the reaction system (including the catalyst surface and the solution).

## 2.5. Analytical methods

The concentration of SMT was analysed by using a HPLC (LC-20AD, Shimadzu) equipped with a diode array detector. Agilent ZORBAX SB-Aq column (5 µm, 4.6 × 250 mm) was used for the separation of SMT. The injected volume was 20 µl at a flow rate of $1 \, ml \, min^{-1}$ and the column temperature was at 25°C. The mobile phase was a mixture of 0.1% formic acid and acetonitrile (81 : 19, v/v). The concentrations of $H_2O_2$ and iron ion were measured using the titanium sulfate method and o-phenanthroline method, respectively [42,43]. The TOC content was measured by using Shimadzu TOC-5000. The intermediate products were identified by using ultra-performance liquid chromatography/tandom mass spectrometry (UPLC-MS) system (G2-XS QTof, Waters) with an ACQUITY UPLC BEH C18 column (1.7 µm, 2.1 × 100 mm). The injected volume was 2 µl, and the flow rate was $0.4 \, ml \, min^{-1}$. The mobile phase A consisted of 0.1% formic acid in water, and the mobile phase B consisted of 0.1% formic acid in acetonitrile. The gradient programme was used: (i) 5% B for the first 2 min; (ii) B was linearly increased to 95% from 2 to 17 min; and (iii) 95% B was held until 19 min. The MS was performed with a selected

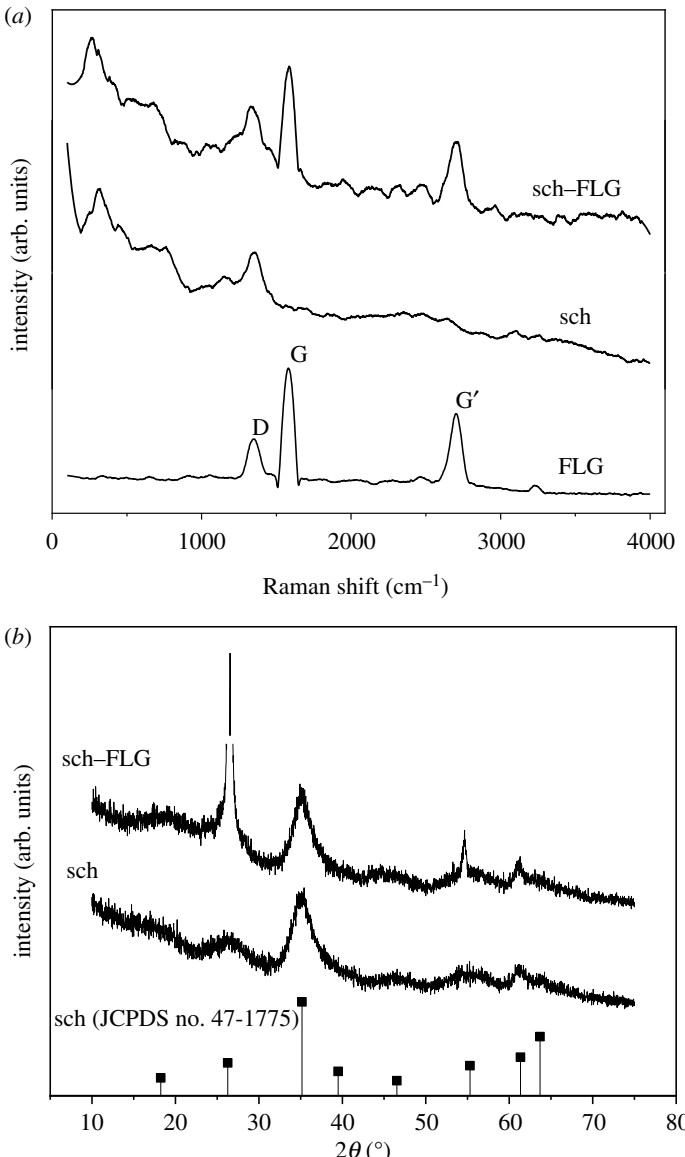

**Figure 1.** (*a*) Raman spectra of FLG, sch and sch–FLG, and (*b*) XRD of sch and sch–FLG.

mass mode (50–1200 *m/z*), using an electrospray source in positive ion mode. The other MS parameters were as follows: the capillary voltage was 3.0 kV, cone voltage was 40 V, source temperature was 120°C and desolvation gas temperature was 400°C.

# 3. Results and discussion

## 3.1. Characterization of sch/few-layer graphene composite

As shown in figure 1*a*, the Raman spectrum of FLG shows peaks G and G′ of graphene at 1582 and 2700 cm$^{-1}$, which is similar to the Raman spectrum of three-layer graphene. The D peak at 1350 cm$^{-1}$ indicates that the graphene material has more edges and flaws. The D, G and G′ peaks on the spectrum of the FLG can be identified in the sch–FLG, and the broad peak whose Raman shift is less than 1582 cm$^{-1}$ corresponds to the Raman spectrum of the sch. Therefore, the sch–FLG is composed of sch and FLG [44,45].

The XRD patterns of sch–FLG and sch are shown in figure 1*b*. The peak at 26.48° shown in the pattern of sch–FLG was recognized as (002) reflection of FLG [46]. Seven broad peaks (2$\theta$ = 18.24, 26.27, 35.16, 39.49, 46.53, 55.29, 61.34°) shown in the patterns of sch and sch–FLG matched well with the standard

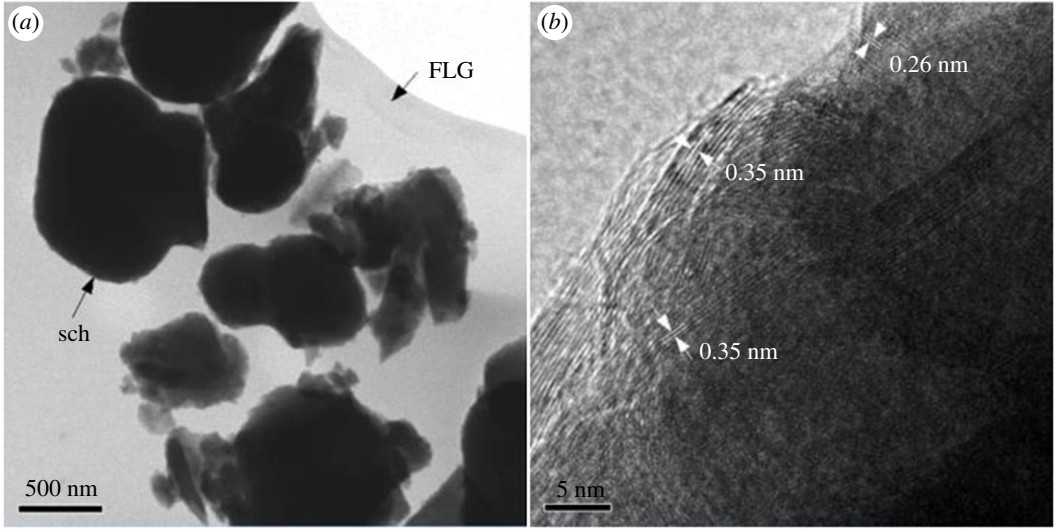

**Figure 2.** TEM images of schwertmannite/few-layer graphene composite: (*a*) ×2000 and (*b*) ×600 000.

diffraction data for sch (JCPDS no. 47-1775) [47]. These results suggest the crystalline structure of sch carried by FLG was not obviously changed during the synthesis of sch–FLG.

Figure 3 shows the HRTEM micrographs of sch–FLG at different magnification levels. It can be seen from figure 2*a* that sch particles are distributed in the film-like structure of FLG (figure 2*a*). The diameter sizes of sch particles were about 500 nm, matching with the values reported by other studies [48,49]. The specific surface area of sch–FLG was much higher than that of sch (5.4 $m^2 g^{-1}$ versus 2.08 $m^2 g^{-1}$). As shown in figure 2*b*, the lattice fringe spacing of 0.26 and 0.35 nm corresponded to the reflection of (212) and (310) planes of sch. Thus, the results of XRD and HRTEM analysis clearly reveal that sch was successfully carried by FLG in sch–FLG composite.

The chemical bonding states on the surface of sch–FLG were characterized by XPS. As shown in figure 3*a*, the O element in sch was mostly from $SO_4^{2-}$ (531.5 eV), Fe–OH (532.0 eV) and Fe–O (530.1 eV) [50–52]. When sch was carried by FLG, new bonds of Fe–O–C (531.2 eV), C–OH and C–O–C (533.0 eV) appeared [53,54]. It was reported that the graphene can bond with iron oxides through the Fe–O or Fe–O–C bond [55,56], and the electrical conductivity can be enhanced by the Fe–O–C bond between graphene and iron oxide to accelerate the oxidation and reduction progresses taking place on the surfaces of catalysts [57,58]. In the present study, although the bonds of O–C=O (289.2 eV), C–OH or C–O–C (285.3 eV), C–C (284.8 eV), and C=C (531.5 eV) were observed on the surface of sch–FLG (figure 3*b*), the Fe–C bond was not observed. These results suggest that sch was connected with FLG mainly via the chemical bond of Fe–O–C on the surface of sch–FLG.

## 3.2. Catalytic activity of schwertmannite/few-layer graphene composite in a heterogeneous Fenton-like reaction

The degradation of SMT with reaction time was studied in the Fenton-like reactions catalysed by sch–FLG, FLG, and sch. As shown in figure 4*a*, almost no removal of SMT was observed when $H_2O_2$ solution was added alone, which indicates that $H_2O_2$ alone cannot degrade SMT. Less than 16.1% of SMT was degraded in 180 min by the Fenton-like reactions catalysed by 0.13 g $l^{-1}$ of FLG which is equal to the amount of FLG in 1 g $l^{-1}$ of sch–FLG. When 1 g $l^{-1}$ of sch or sch–FLG was used to catalyze the heterogeneous Fenton-like reaction, 27.6% and 100% of SMT was degraded in 120 min, respectively. Obviously, compared to sch or FLG, sch–FLG was more effective to catalyse the heterogeneous Fenton-like reaction to degrade SMT. Thus, sch–FLG exhibited a much higher catalytic activity than sch or FLG for the degradation of SMT in the heterogeneous Fenton-like reaction.

In heterogeneous Fenton-like processes, the reaction parameters, such as $H_2O_2$ concentration, catalyst dosage, initial solution pH and reaction temperature, can greatly influence the degradation efficiency or organic contaminants [59,60], and thus the influences of these parameters in the Fenton-like reaction catalysed by sch–FLG were investigated. The effect of $H_2O_2$ dosage on the degradation of SMT during a Fenton-like reaction catalysed by sch–FLG is shown in figure 4*b*. Less than 7.6% of SMT was

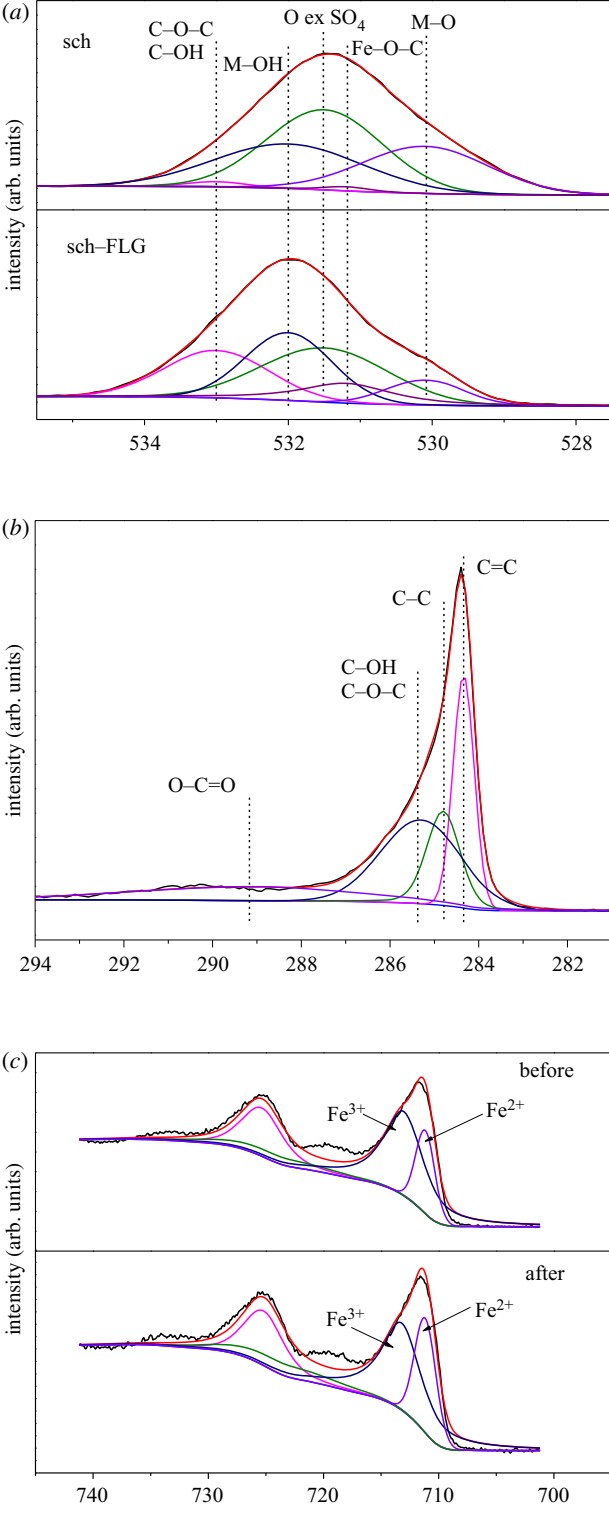

**Figure 3.** XPS of (*a*) O 1s for sch (sch) and sch/FLG composite (sch–FLG), (*b*) C 1s for sch–FLG and (*c*) Fe 2p for sch–FLG before and after use.

removed when only $1\,g\,l^{-1}$ of sch–FLG was added (without the addition of $H_2O_2$), indicating that the adsorption of sch–FLG for SMT was very low. By loading $100\,mg\,l^{-1}$ $H_2O_2$, 95.64% of SMT was degraded in 180 min. The degradation efficiency of SMT can be further increased via increasing the dosage of $H_2O_2$ to $200–500\,mg\,l^{-1}$. For instance, SMT can be completely removed from the solution in only 90 min when 200 or $500\,mg\,l^{-1}$ $H_2O_2$ was loaded. However, the time required for the complete removal of SMT was prolonged to 120 min when further increasing the load of $H_2O_2$ to $1000\,mg\,l^{-1}$,

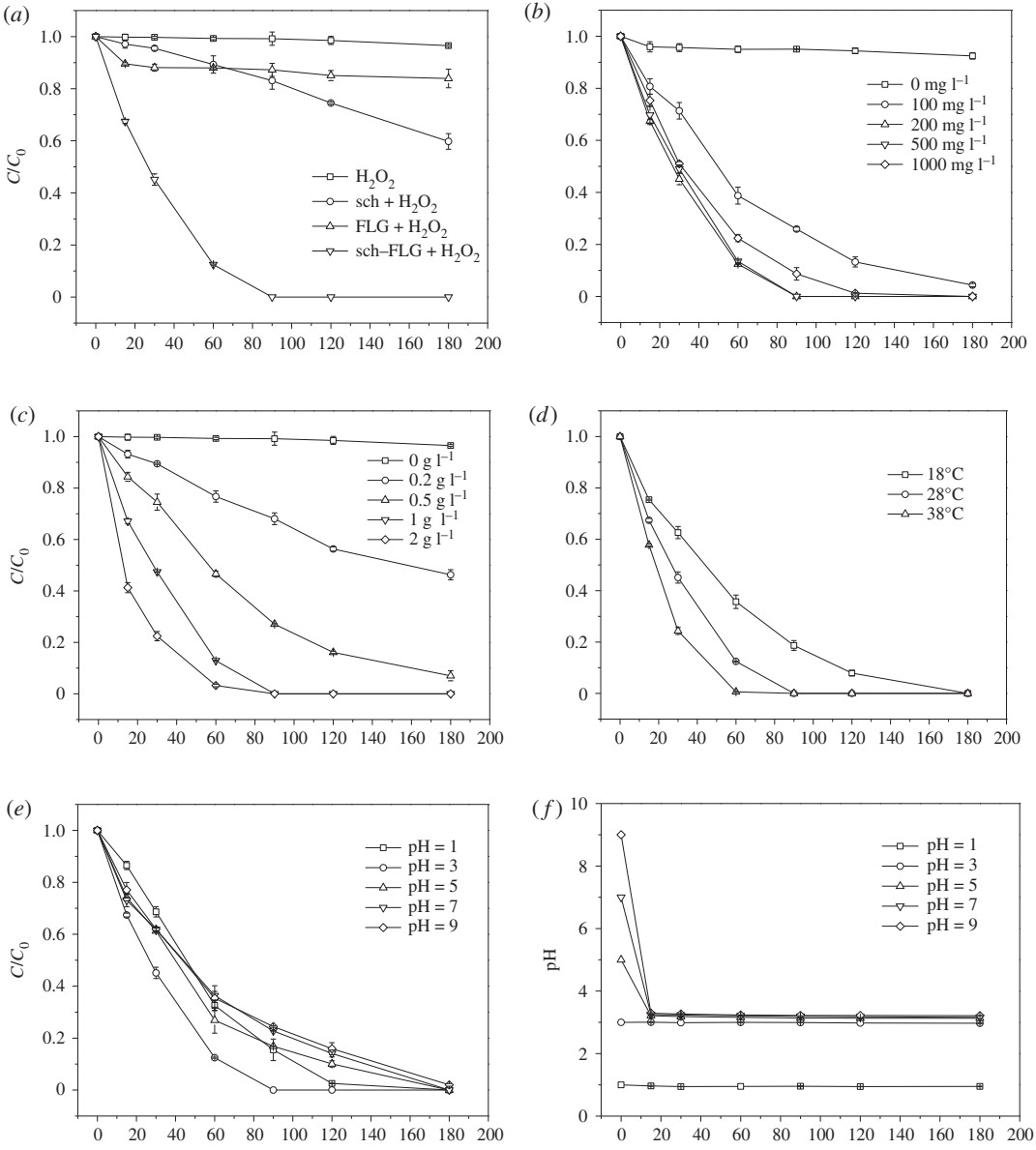

**Figure 4.** Effect of (*a*) different catalysts, (*b*) $H_2O_2$ dosage, (*c*) catalyst dosage, (*d*) reaction temperature, and (*e*) solution initial pH on the degradation of SMT. (*f*) Effect of initial solution pH on the pH evolution during the reaction. Experimental conditions: (*a*) SMT 5 mg $l^{-1}$, $H_2O_2$ 200 mg $l^{-1}$, catalyst dosage 1 g $l^{-1}$ except FLG 0.13 g $l^{-1}$, pH 3.0, temperature 28°C; (*b*) SMT 5 mg $l^{-1}$, sch–FLG 1 g $l^{-1}$, pH 3.0, temperature 28°C; (*c*) SMT 5 mg $l^{-1}$, $H_2O_2$ 200 mg $l^{-1}$, pH 3.0, temperature 28°C; (*d*) SMT 5 mg $l^{-1}$, sch–FLG 1 g $l^{-1}$, $H_2O_2$ 200 mg $l^{-1}$, pH 3.0; (*e*) and (*f*) SMT 5 mg $l^{-1}$, sch–FLG 1 g $l^{-1}$, $H_2O_2$ 200 mg $l^{-1}$, temperature 28°C.

most probably owing to the fact that excessive $H_2O_2$ (1000 mg $l^{-1}$) in the solution would capture –OH to form HO 2· to lower the degradation efficiency of SMT [61,62].

As shown in figure 4*c*, when the dosage of sch–FLG was raised from 0.2 to 1 g $l^{-1}$, the degradation efficiency of SMT in 90 min increased from 31.93% to 100%. However, the time required for the complete removal of SMT was not further shortened when increasing its dosage 2 g $l^{-1}$. It can thus be inferred that although higher dosage of sch–FLG provided more active sites to generate –OH, excessive iron species would inhibit the degradation of SMT owing to the consumption of –OH by $Fe^{2+}$ [63]. The effect of reaction temperature on the degradation of SMT during the Fenton-like reaction catalysed by sch–FLG is shown in figure 4*d*. The degradation efficiency of SMT was increased when the reaction temperature was raised from 18°C to 38°C, and SMT can be completely removed from the solution in only 60 min at 38°C. In fact, previous studies also reported that within a certain range of temperatures, higher temperature can accelerate the oxidation and reduction reaction between Fe(II) and Fe(III) to promote the generation of –OH, thus increasing the degradation efficiency of organic contaminant [64,65].

At the initial solution pH of 3.0, SMT can be removed from the solution in 90 min (figure 4*e*). When the initial solution pH was 1.0, sch–FLG was dissolved in the solution to release the iron ions, thus activating the homogeneous Fenton-like process to degrade 98.5% of SMT in 90 min. The degradation efficiency of SMT in 90 min was still as high as 75.5% when the initial solution was increased to 9.0. These results indicated that sch–FLG can adapt to a wide range of initial solution pH. To reveal why sch–FLG has such outstanding adaptability for the initial solution pH, the change in solution pH during the reactions was recorded and is shown in figure 4*f*. When the initial solution pH was higher than 3, the solution pH decreased to around 3 in the first 15 min. Clearly, the sch–FLG can balance the solution pH to accelerate the Fenton-like reaction catalysed by sch–FLG. On one hand, sch has plenty of sulfate adsorbed in its outer sphere, the dissolution of which can cause the release of H$^+$ from the surface of sch (equation (3.1)) [66]. On the other hand, iron oxides can adsorb H$_2$O molecules, form an OH− complex with surface iron (≡FeOH), and dissolve H$^+$ into the solution when they are introduced into water [67]. Given the fact that the point of zero charge pH (pHpzc) of sch was 3.05 [67,68], the solution pH would decrease through equation (3.2) when it was higher than the pHpzc of sch [12,14]. In summary, the decrease of solution pH during the Fenton-like reaction catalysed by sch–FLG most probably resulted from the above two processes.

The performance of the catalysts in some other studies is shown in table 1. Those catalytic materials generally need to be in a higher temperature (35–45°C) and a narrower pH (2–3.5) range in the catalytic degradation of sulfamethoxazole [65,69,70]. However, we found that the catalytic degradation efficiency of SMT (5 g l$^{-1}$) by sch–FLG was around 100% at a lower temperature (28°C), and in a wide range of initial solution pH values (1–9). It can be seen that sch–FLG has excellent catalytic performance and adapts to a wider pH range:

$$\equiv FeOH_2^+ SO_4^{2-} \rightarrow \equiv FeOH + SO_4^{2-} + H^+ \tag{3.1}$$

and

$$\equiv FeOH + OH^- \rightarrow \equiv FeO^- + H_2O. \tag{3.2}$$

## 3.3. Identification of reactive oxidizing species

To identify the main reactive oxidizing species in the Fenton-like system catalysed by sch–FLG, KI and methanol were respectively added to scavenge the –OH on the surface of sch–FLG and in the whole reaction system [71,72]. As shown in figure 5, only 8.02% or 4.67% of SMT was removed in 90 min when KI and methanol were respectively added, implying that the main reactive oxidizing species in the Fenton-like reaction is the –OH generated on the surface of sch–FLG. Figure 3*c* shows the Fe 2p high-resolution scan spectra of sch–FLG before and after use. The peak at 725 and 711 eV can be ascribed to Fe 2p$_{1/2}$ and Fe 2p$_{3/2}$. The Fe 2p$_{3/2}$ peak can be deconvoluted into two sub peaks corresponding to Fe(III) (713.2 eV) and Fe(II) (711.2 eV) [51,73]. The intensity ratio of Fe(III)/Fe(II) on the surface of sch–FLG before and after use is 3.03 and 2.14, respectively, revealing that a part of Fe(III) on the surface of sch–FLG was reduced to Fe(II). Thus, the iron on the surface of sch–FLG took part in the oxidation–reduction reaction, and the hydroxyl radicals were mainly generated on the surface of sch–FLG (equations (3.3) and (3.4)). In addition, it can also be inferred from the above results that the FLG as an electron donor–acceptor enhanced the electron conduction rate through the Fe–O–C bond between FLG and sch, thus accelerating the oxidation–reduction reaction to generate –OH and resulting in the much higher catalytic activity of sch–FLG:

$$\equiv Fe(II) + H_2O_2 \rightarrow \equiv Fe(III)-OH + -OH \tag{3.3}$$

and

$$\equiv Fe(III)-OH + H_2O_2 \rightarrow \equiv Fe(II) + HO_2 + H_2O. \tag{3.4}$$

## 3.4. H$_2$O$_2$ and total organic carbon evolution, iron leaching and the reusability of schwertmannite/few-layer graphene

The evolution of H$_2$O$_2$ and TOC during the Fenton-like degradation of SMT catalysed by sch–FLG was determined. As shown in figure 6, the concentration of H$_2$O$_2$ gradually declined from 200 to 25.33 mg l$^{-1}$ and 66.81% of TOC was removed in 24 h of reaction. The utilization efficiency of H$_2$O$_2$ was calculated

**Table 1.** Performance of the catalysts in other studies.

| materials | target pollutant | $C_0$ (mg l$^{-1}$) | $H_2O_2$ (mg l$^{-1}$) | catalyst g l$^{-1}$ | pH | $T$ (°C) | time (min) | removal efficiency (%) | references |
|---|---|---|---|---|---|---|---|---|---|
| Fe$_3$O$_4$/Mn$_3$O$_4$ | sulfamethazine | 20 | 204 | 0.5 | 2.5–3 | 45 | 50 | 100 | [69] |
| Fe$_3$O$_4$/Mn$_3$O$_4$/rGO | | 20 | 285 | 0.5 | 3–3.5 | 35 | 80 | 98 | [65] |
| Fe$_3$O$_4$ magnetic nanoparticles | | 20 | 680 | 1 | 2–3 | — | 150 | 80 | [70] |
| sch–FLG | | 5 | 200 | 1 | 1–9 | 28 | 90 | 100 | this work |

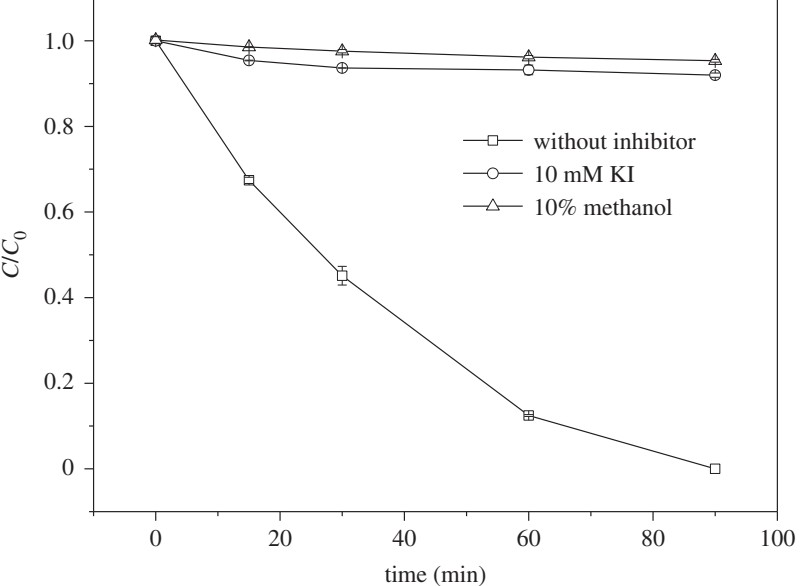

**Figure 5.** Effect of different inhibitors on the degradation of SMT. Experimental conditions: SMT 5 mg $l^{-1}$, $H_2O_2$ 200 mg $l^{-1}$, sch–FLG 1 g $l^{-1}$, pH 3.0, temperature = 28°C.

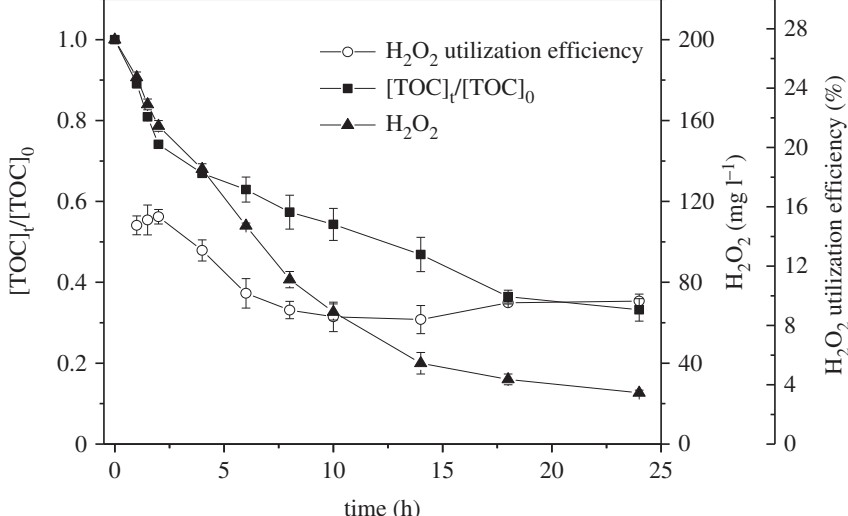

**Figure 6.** Evolution of the concentrations of TOC, $H_2O_2$ and the $H_2O_2$ utilization efficiency in degradation of SMT. Experimental conditions: SMT 5 mg $l^{-1}$, $H_2O_2$ 200 mg $l^{-1}$, sch–FLG 1 g $l^{-1}$, pH 3.0, temperature = 28°C.

through equation (3.5) [74]:

$$\eta(\%) = \frac{k \times [\text{SMT}]}{[\text{H}_2\text{O}_2]_{\text{con}}} \times 100\%, \tag{3.5}$$

where $\eta$ is the utilization efficiency of $H_2O_2$ (%); $k$ is the theoretical stoichiometry of $H_2O_2$ to mineralize one mole SMT ($k = 42$); [SMT] is the amount of SMT corresponding to the TOC mineralized (mM); and $[\text{H}_2\text{O}_2]_{\text{con}}$ is the amount of $H_2O_2$ consumed in the reaction (mM). The highest utilization efficiency of $H_2O_2$ is 15.33% in 2 h, and then it decreased to 9.64% in 24 h.

The leaching of iron ions was monitored during the degradation process. As shown in figure 7, the concentration of total iron in the solution was 1.23 and 2.88 mg $l^{-1}$ at 90 min and 24 h of reaction, respectively, which were only equal to 0.12% and 0.29% of the iron in the used sch–FLG. In addition, the leached iron in the solution almost all comprised $Fe^{3+}$. The reusability of sch–FLG was further assessed through using it for a consecutive five cycles to catalyse the Fenton-like reaction. As shown in figure 8, 87.87% and 100% of SMT was degraded in 80 min and 120 min in the first cycle. In the

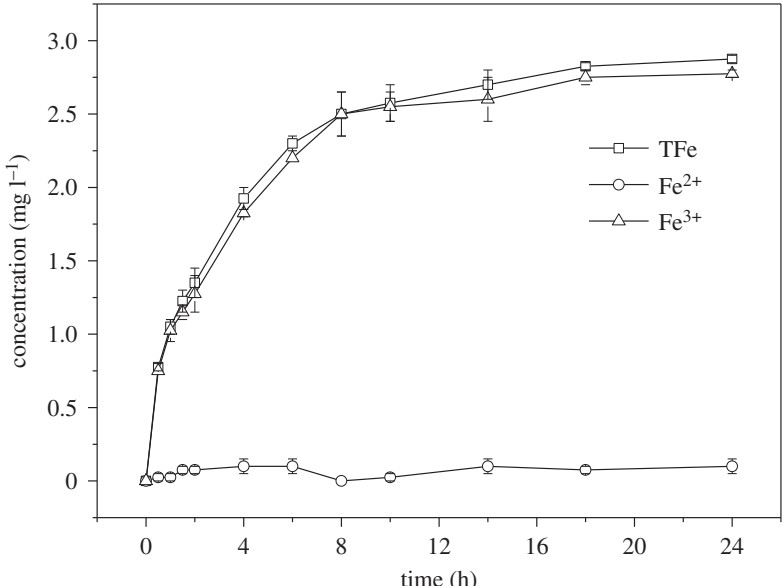

**Figure 7.** The evolution of the iron leaching on the degradation of SMT. Experimental conditions: SMT 5 mg $l^{-1}$, $H_2O_2$ 200 mg $l^{-1}$, sch–FLG 1 g $l^{-1}$, pH 3.0, temperature = 28°C.

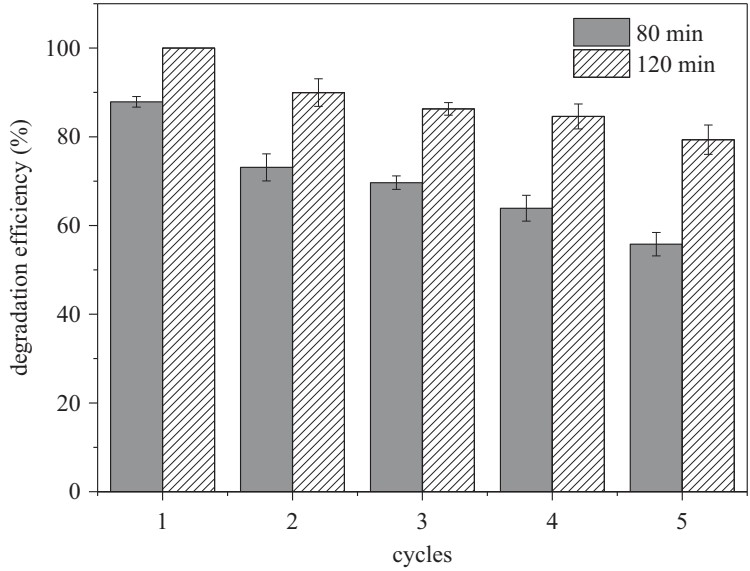

**Figure 8.** Effect of catalyst repeat use on degradation of SMT. Experimental conditions: SMT 5 mg $l^{-1}$, $H_2O_2$ 200 mg $l^{-1}$, sch–FLG 1 g $l^{-1}$, pH 3.0, temperature = 28°C.

next four cycles, the degradation efficiency of SMT in 80 min was in the range of 55.81%–73.10%, and the degradation efficiency of SMT in 120 min ranged from 79.35% to 89.96%. Compared with the pristine sch–FLG, there was no obvious change on the XRD pattern of the repeatedly used sch–FLG (electronic supplementary material, figure S2). Thus, the sch–FLG can maintain a certain catalytic activity for the degradation of SMT, and its mineral structure was not changed during its repeated uses in a Fenton-like reaction.

## 3.5. Possible degradation pathways of sulfamethazine

The intermediate products involved in the degradation of SMT were identified (table 2) and the possible degradation pathways of SMT were proposed (figure 9). When the aromatic ring or the R-substituent group was attached to the amide group of sulfonamides, the strong electrophilic addition of hydroxyl

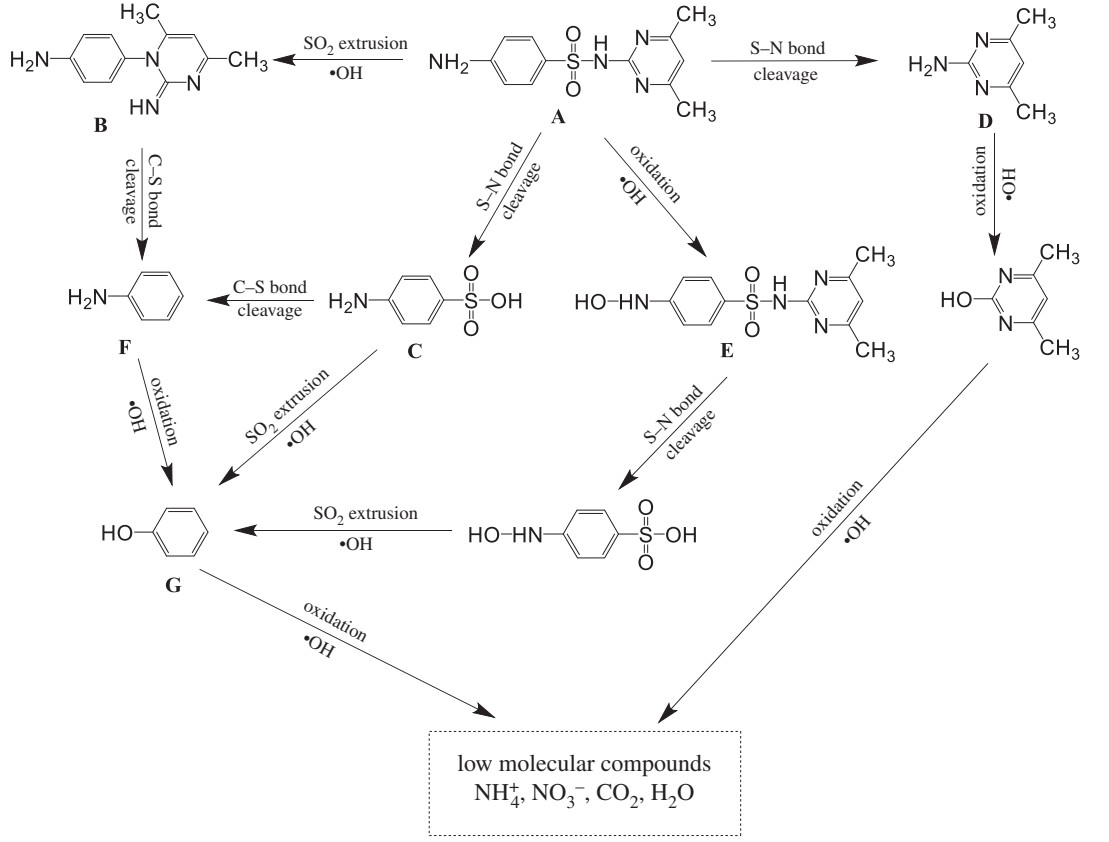

**Figure 9.** The possible pathways of SMT degradation catalysed by sch–FLG.

**Table 2.** The main intermediate products identified during the SMT degradation catalysed by sch–FLG.

| | products | formula | molecular structure | m/z |
|---|---|---|---|---|
| A | sulfamethazine | $C_{12}H_{14}O_2N_4S$ | | 279 |
| B | 4-(2-imino-4,6-dimethylpyrimidin-1(2H)-yl) aniline | $C_{12}H_{14}N_4$ | | 215 |
| C | sulfanilic acid | $C_6H_7O_3NS$ | | 174 |
| D | 4,6-dimethylpyrimidin-2-amine | $C_6H_9N_3$ | | 124 |
| E | hydroxylated sulfamethazine | $C_{12}H_{14}O_3N_4S$ | | 295 |
| F | aniline | $C_6H_7N$ | | 94 |
| G | phenol | $C_6H_6O$ | | 95 |

radicals would make SMT hydroxylate. As a result, the hydroxylated SMT was identified as an intermediate product [75]. By the cleavage of the S–N bond, the hydroxylated SMT might be further broken into 4-(hydroxyamino) benzenesulfonic acid, which can be degraded to phenol by the $SO_2$ extrusion and –OH oxidation. In this process, the $NO_3^-$ was formed by the oxidation of the N atom attached to the aromatic rings and the $SO_4^{2-}$ was released to solution.

The $SO_2$ extrusion of SMT formed 4-(2-imino-4, 6-dimethylpyrimidin-1(2H)-yl) aniline, which can be broken into aniline and 6-dimethylpyrimidin-2-amine by C–N bound cleavage [76]. The cleavage of the S–N bond broke SMT into sulfanilic acid and 4,6-dimethylpyrimidin-2-amine. The sulfanilic acid might be further degraded to aniline by C–S bond cleavage and phenol by C–N bond cleavage, respectively. 4,6-dimethylpyrimidin-2-amine would be destroyed by the cleavage of C=N and C–N bonds on the pyrimidine ring and be oxidized to low molecular compounds by –OH [77]. The aniline would be oxidized into phenol that can be easily oxidized into $CO_2$ and $H_2O$.

# 4. Conclusion

In the present study, sch–FLG was synthesized in order to promote the catalytic activity of sch in a heterogeneous Fenton-like reaction. Results showed that sch can be successfully carried by FLG in sch–FLG composite, mainly via the chemical bond of Fe–O–C on the surface of sch–FLG. The sch–FLG exhibited a much higher catalytic activity than sch or FLG for the degradation of SMT in the heterogeneous Fenton-like reaction. The degradation efficiency of SMT was around 100% under the reaction conditions of $H_2O_2$ 200–500 mg $l^{-1}$, sch–FLG dosage 1–2 g $l^{-1}$, temperature 28–38°C, and initial solution of pH 1–9. The main reactive oxidizing species in the Fenton-like reaction catalysed by sch–FLG is the –OH generated on the surface of sch–FLG. During the repeated uses of sch–FLG in the Fenton-like reaction, it can maintain a certain catalytic activity for the degradation of SMT and the mineral structure was not changed, suggesting a good reusability. In addition, SMT can be finally mineralized in the Fenton-like reaction catalysed by sch–FLG, and possible degradation pathways were proposed. Therefore, the sch–FLG is an excellent catalyst for SMT degradation in a heterogeneous Fenton-like reaction.

Data accessibility. Data are available from the Dryad Digital Repository: https://doi.org/10.5061/dryad.8931zcrm6 [78].

Authors' contributions. D.W contributed to the conception of the study, coordinated the study and helped draft the manuscript; Y.G. and Z.Y. performed the experiments and participated in data analysis; L.Z. helped perform the analysis with constructive discussions.

Competing interests. We have no competing interests.

Funding. The work was supported by the Jiangsu Agriculture Science and Technology Innovation Fund (JASTIF) CX(17)2024 and the National Natural Science Foundation of China (grant nos 41977338, 21637003).

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
