## [Reviewer comments · Royal Society Open Science]

Review History

RSOS-191977.R0 (Original submission)

Review form: Reviewer 1 (Dattatray Late)

Is the manuscript scientifically sound in its present form?

Yes

Are the interpretations and conclusions justified by the results?

Yes

Is the language acceptable?

No

Do you have any ethical concerns with this paper?

No

Have you any concerns about statistical analyses in this paper?

No

Recommendation?

Accept with minor revision (please list in comments)

Comments to the Author(s)

Review attached (Appendix A).

Review form: Reviewer 2**Is the manuscript scientifically sound in its present form?**

Yes

Are the interpretations and conclusions justified by the results?

Yes

Is the language acceptable?

Yes

Do you have any ethical concerns with this paper?

No

Have you any concerns about statistical analyses in this paper?

No

Recommendation?

Major revision is needed (please make suggestions in comments)

Comments to the Author(s)

The article presents an interesting proposal, some changes are required.

The meaning of SMT must be included in the abstract.

The word "weigh" is incorrect.

"ml" must be changed to "mL".

".OH" must be changed to "-OH".

Figure 1 must be completely indexed.

Notation used in line 43 of the page 13 must be verified.

Equation editor should be used to edit chemical reactions.

Equation editor should be used to edit ions in different parts of the text.

It is important to add a table to compare the results of this article with others published with different approaches.

The names of reference 2 should be reviewed.

The names of reference 9 should be reviewed.

The names of reference 16 should be reviewed.

The names of reference 24 should be reviewed.

The names of reference 25 should be reviewed.

The names of reference 28 should be reviewed.

Reference 33 is incorrect.

Reference 34 is incomplete.

Reference 43, doi is incorrect.

The names of reference 51 should be reviewed.

The names of reference 55 should be reviewed.

Reference 63 must use subindices.

Reference 65 must be reviewed.

Decision letter (RSOS-191977.R0)

13-Jan-2020

Dear Dr Wang:

Title: Synthesis and assessment of schwertmannite/few-layer graphene composite for the degradation of sulfamethazine in heterogeneous Fenton-like reaction
Manuscript ID: RSOS-191977

The editor assigned to your manuscript has now received comments from reviewers. We would like you to revise your paper in accordance with the referee and Subject Editor suggestions which can be found below (not including confidential reports to the Editor). Please note this decision does not guarantee eventual acceptance.

Please submit your revised paper before 05-Feb-2020. Please note that the revision deadline will expire at 00.00am on this date. If we do not hear from you within this time then it will be assumed that the paper has been withdrawn. In exceptional circumstances, extensions may be possible if agreed with the Editorial Office in advance. We do not allow multiple rounds of revision so we urge you to make every effort to fully address all of the comments at this stage. If deemed necessary by the Editors, your manuscript will be sent back to one or more of the original reviewers for assessment. If the original reviewers are not available we may invite new reviewers.

RSC Associate Editor:
Comments to the Author:
(There are no comments.)

RSC Subject Editor:
Comments to the Author:
(There are no comments.)

Reviewers' Comments to Author:
Reviewer: 1

Comments to the Author(s)
Review attached

Reviewer: 2

Comments to the Author(s)
The article presents an interesting proposal, some changes are required.

The meaning of SMT must be included in the abstract.

The word "weigh" is incorrect.

"ml" must be changed to "mL".

".OH" must be changed to "-OH".

Figure 1 must be completely indexed.

Notation used in line 43 of the page 13 must be verified.

Equation editor should be used to edit chemical reactions.

Equation editor should be used to edit ions in different parts of the text.

It is important to add a table to compare the results of this article with others published with different approaches.

The names of reference 2 should be reviewed.

The names of reference 9 should be reviewed.

The names of reference 16 should be reviewed.

The names of reference 24 should be reviewed.

The names of reference 25 should be reviewed.

The names of reference 28 should be reviewed.

Reference 33 is incorrect.

Reference 34 is incomplete.

Reference 43, doi is incorrect.

The names of reference 51 should be reviewed.

The names of reference 55 should be reviewed.

Reference 63 must use subindices.

Reference 65 must be reviewed.

Author's Response to Decision Letter for (RSOS-191977.R0)

See Appendix B.

RSOS-191977.R1 (Revision)

Review form: Reviewer 2

Is the manuscript scientifically sound in its present form?

Yes

Are the interpretations and conclusions justified by the results?

Yes

Is the language acceptable?

Yes

Do you have any ethical concerns with this paper?

No

Have you any concerns about statistical analyses in this paper?

No

Recommendation?

Accept with minor revision (please list in comments)

Comments to the Author(s)

In reference 2, the use of accents and umlauts is not trivial in the names of the authors.

In reference 16, the use of hyphen is not trivial in the names of the authors.

In reference 26, the last name must be Diniz da Costa JC.

In reference 37, the use of accents must be using microsoft word not using images or other font type.

In references 69 and 70, the authors change the format of separating a name in two parts that they had used previously.

Decision letter (RSOS-191977.R1)

Dear Dr Wang:

Title: Synthesis and assessment of schwertmannite/few-layer graphene composite for the degradation of sulfamethazine in heterogeneous Fenton-like reaction

Manuscript ID: RSOS-191977.R1

Thank you for submitting the above manuscript to Royal Society Open Science. On behalf of the Editors and the Royal Society of Chemistry, I am pleased to inform you that your manuscript will be accepted for publication in Royal Society Open Science subject to minor revision in accordance with the referee suggestions. Please find the reviewers' comments at the end of this email.

The reviewers and handling editors have recommended publication, but also suggest some minor revisions to your manuscript. Therefore, I invite you to respond to the comments and revise your manuscript.

Because the schedule for publication is very tight, it is a condition of publication that you submit the revised version of your manuscript before 21-Jun-2020. Please note that the revision deadline will expire at 00.00am on this date. If you do not think you will be able to meet this date please let me know immediately.

Kind regards,
Dr Laura Smith
Publishing Editor, Journals

Royal Society of Chemistry
Thomas Graham House

Science Park, Milton Road
Cambridge, CB4 0WF
Royal Society Open Science - Chemistry Editorial Office

RSC Associate Editor:
Comments to the Author:
(There are no comments.)

RSC Subject Editor:
Comments to the Author:
(There are no comments.)

Reviewer comments to Author:
Reviewer: 2

Comments to the Author(s)
In reference 2, the use of accents and umlauts is not trivial in the names of the authors.
In reference 16, the use of hyphen is not trivial in the names of the authors.
In reference 26, the last name must be Diniz da Costa JC.
In reference 37, the use of accents must be using microsoft word not using images or other font type.
In references 69 and 70, the authors change the format of separating a name in two parts that they had used previously.

Author's Response to Decision Letter for (RSOS-191977.R1)

See Appendix C.

Decision letter (RSOS-191977.R2)

Dear Dr Wang:

Title: Synthesis and assessment of schwertmannite/few-layer graphene composite for the degradation of sulfamethazine in heterogeneous Fenton-like reaction
Manuscript ID: RSOS-191977.R2

It is a pleasure to accept your manuscript in its current form for publication in Royal Society Open Science. The chemistry content of Royal Society Open Science is published in collaboration with the Royal Society of Chemistry.

RSC Associate Editor
Comments to the Author:
(There are no comments.)

Reviewer(s)' Comments to Author:

Appendix A

Comment on # RSOS-191977

The present investigations reports the sch-FLG was synthesized in order to promote the catalytic activity of sch in heterogeneous Fenton-like reaction. Results showed that schwertmannite can be successfully carried by FLG in sch-FLG composite, mainly via the chemical bond of Fe-O-C on the surface of sch-FLG. The sch-FLG exhibited a much higher catalytic activity than schwertmannite or FLG for the degradation of SMT in the heterogeneous Fenton-like reaction. The degradation efficiency of SMT was around 100% under the reaction conditions of H₂O₂ 200-500 mg L⁻¹, sch-FLG dosage 1-2 g L⁻¹, temperature 28-38 °C, and initial solution pH 1-9. The main reactive oxidizing species in the Fenton-like reaction catalyzed by sch-FLG is the ·OH generated on the surface of sch-FLG. During the repeated uses of sch-FLG in Fenton-like reaction, it can maintain a certain catalytic activity for the degradation of SMT and the mineral structure was not changed during, suggesting a good reusability. I recommend publication of this manuscript after appropriate revision.

1. What is the importance of sch-FLG mixed-dimensional structure?
 2. Raman spectroscopy data of XRD of schwertmannite (sch) and schwertmannite/few-layer graphene composite (sch-FLG) need to be added.
 3. Materials characterizations using XPS were given but and its detail analysis need to be given.
 4. What is the cyclic stability of the catalyst material ?
1. Some relevant literature need to be added on For example Materials Research Express 6 (2019), 095066; Journal of Physics and Chemistry of Solids 128, (2019) 384-390; Applied Physics Letters 107 (2015), 123503; Physica Status Solidi (a) 216 (2019) 1900121; Applied Physics Letters 106 (2015), 023111; Journal of the Science of Food and Agriculture 95 (2015), 2772–2778; Appl. Phys. Lett. 105 (2014), 243502; Journal of Experimental Nanoscience 6 (2011), 641-651; Journal of Physics: Condensed Matter 23 (2011), 055303; Journal of Experimental Nanoscience 4 (2009), 313-322

Appendix B

Manuscript number: RSOS-191977

Title: Synthesis and assessment of schwertmannite/few-layer graphene composite for the degradation of sulfamethazine in heterogeneous Fenton-like reaction

Thanks the editor and reviewers for your critical review of our manuscript with constructive and valuable comments. The itemized responds to the Editor and reviewer's comments and the main corrections in the revised version are as flowing:

1. What is the importance of sch-FLG mixed-dimensional structure?

Reply: Thank you very much for your comment. The importance of sch-FLG mixed-dimensional structure lies in (1) the specific surface area of sch-FLG was much higher than that of schwertmannite, the mixed-dimensional structure can increase the specific surface area of the material to get more reaction point; (2) on the surface of sch-FLG, schwertmannite was connected with FLG mainly via Fe-O-C bond the graphene can serve as electron donor-acceptor to enhance the conduction of electron, thus accelerating the oxidation and reduction reactions on the surface of catalyst.

2. Raman spectroscopy data of XRD of schwertmannite (sch) and schwertmannite/few-layer graphene composite (sch-FLG) need to be added.

Reply: Thank you very much for your thoughtful and constructive comments. According to the preparation method mentioned in the original paper, we carried out

to chemically synthesize schwertmannite (sch) and schwertmannite/few-layer graphene composite (sch-FLG) again. Then few-layer graphene (FLG), sch and sch-FLG were detected by laser Raman spectrometer (HR Evolution, HORIBA FRANCE SAS) in a spectrum scanning range of 100-4000 cm^{-1} using a solid state semiconductor laser with $\lambda = 532 \text{ nm}$. The results are shown in the figure 1.

Figure 1. Raman spectra of FLG, sch and sch-FLG

As shown in Figure 1, the Raman spectrum of FLG shows peak G and G' of graphene at 1582 cm^{-1} and 2700 cm^{-1} , which is similar to the Raman spectrum of 3-layer graphene. And the D peak at 1350 cm^{-1} indicates that the graphene material has more edges and flaws. The D, G and G' peak on the spectrum of the FLG can be identified in the sch-FLG, and the broad peak whose Raman shift is less than 1582 cm^{-1} corresponds to the Raman spectrum of the sch. Therefore, the sch-FLG is composed of sch and FLG (*Late et al., J Phys: Condens Matter, 2011, 23, 055303*;

Paton et al., Nat mater, 2014, 13, 624)

In the revised version, these have been added and the Raman and XRD images are combined into one picture. Accordingly, “Fig. 1. XRD of schwertmannite (sch) and schwertmannite/few-layer graphene composite (sch-FLG)” is adjusted to “Fig. 1. (a) Raman spectra of sch, FLG; and sch-FLG, (b) XRD of sch and sch-FLG” in revised version of Figure and Table.

3. Materials characterizations using XPS were given but and its detail analysis need to be given.

Reply: Thank you very much for your comment. Actually, to figure out the mixed-dimensional structure of sch-FLG, we have analyzed the O 1s, C 1s detail data of XPS, and to figure out the confirm the oxidation and reduction on the surface of catalyst, we have analyzed the Fe 2p detail data of XPS.

As shown in figure 2a, the O element in schwertmannite was mostly from SO₂₋₄ (531.5 eV), Fe-OH (532.0 eV) and Fe-O (530.1 eV) (*Zhang et al., Environ Sci Technol., 2005, 39, 7246-7253; Hu et al., Appl Catal B., 2011, 107, 274-283; Geng et al., J Mater Chem., 2012, 22, 3527-3535*). When schwertmannite was carried by FLG, new bond of Fe-O-C (531.2 eV), C-OH and C-O-C (533.0 eV) appeared (*Fan, et al., ACS Nano., 2011, 5, 191-198.43; Chen et al., Adv Mater., 2011, 23: 5679-5683*). It was reported that the graphene can bond with iron oxides through the Fe-O or Fe-O-C bond (*Kataby et al., Langmuir, 1999, 15, 1703-1708; Lin et al., Environ Sci Technol., 1998, 32, 1417-1423*), and the electrical conductivity can be enhanced by the Fe-O-C

bond between graphene and iron oxide to accelerate the oxidation and reduction progresses taking place on the surfaces of catalysts (*Jasuja J Phys Chem Lett.*, 2010, 1, 1853-1860; *Zubir et al., Sci Rep.*, 2014, 4, 4594). In the present study, although the bonds of O-C=O (289.2eV), C-OH or C-O-C (285.3 eV), C-C (284.8 eV), and C=C (531.5 eV) were observed on the surface of sch-FLG (figure 2b), the Fe-C bond was not observed. These results suggest that schwertmannite was connected with FLG mainly via the chemical bond of Fe-O-C on the surface of sch-FLG.

Figure 2c shows the Fe 2p high resolution scan spectra of sch-FLG before and after use. The peak at 725 eV and 711 eV can be ascribed to Fe 2p_{1/2} and Fe 2p_{3/2}. The Fe 2p_{3/2} peak can be deconvoluted into two sub peaks corresponding to Fe(III) (713.2 eV) and Fe(II) (711.2 eV) (*Hu et al., Appl Catal B.*, 2011, 107, 274-283; *Teng et al., Nano Lett.*, 2003, 3, 261-264). The intensity ratio of Fe(III)/Fe(II) on the surface of sch-FLG before and after use is 3.03 and 2.14, respectively, revealing that a part of Fe(III) on the surface of sch-FLG was reduced to Fe(II). Thus, the iron on the surface of sch-FLG took part in the oxidation-reduction reaction, and the hydroxyl radicals were mainly generated on the surface of sch-FLG.

Figure 1. XPS of (a) O 1s for schwertmannite (sch) and schwertmannite/few-layer graphene composite (sch-FLG), (b) C 1s for schwertmannite/few-layer graphene composite, and (c) Fe 2p for schwertmannite/few-layer graphene composite before and after use.

These contents have been expressed in lines 48-60 on page 10, lines 1-25 on page 11, lines 48-60 on page 14, and lines 1-12 on page 15 in the original submission.

4. What is the cyclic stability of the catalyst material?

Reply: Thank you very much for your comment. The cyclic stability of the catalyst material is reflected in two aspects. Firstly, the mineral structure of the catalyst material (sch-FLG) is of high stability during its repeated uses in Fenton-like reaction. Very few iron ions were leached from the catalyst during each reuse process; there was no obvious change on its mineral structure before and after use. Secondly, the sch-FLG can maintain a certain catalytic activity for the degradation of SMT

during its repeated uses.

5. Some relevant literature need to be added on. For example Materials Research Express 6 (2019), 095066; Journal of Physics and Chemistry of Solids 128, (2019) 384-390; Applied Physics Letters 107 (2015), 123503; Physica Status Solidi (a) 216 (2019) 1900121; Applied Physics Letters 106 (2015), 023111; Journal of the Science of Food and Agriculture 95 (2015), 2772–2778; Appl. Phys. Lett. 105 (2014), 243502; Journal of Experimental Nanoscience 6 (2011), 641-651; Journal of Physics: Condensed Matter 23 (2011), 055303; Journal of Experimental Nanoscience 4 (2009), 313-322

Reply: Thank you very much for your suggestions. We have studied these references carefully, which help to improve the quality of our research papers. These relevant literatures have been added to the revised version and all the serial numbers of references have been adjusted accordingly.

6. In addition, Reviewer 2 suggested that it is important to add a table to compare the results of this article with others published with different approaches.

Reply: Thank you very much for your thoughtful and constructive comments. We have made corresponding improvements in the revised paper and added Table 1 to the revised figures and tables. As shown in table 1, those catalytic materials generally need to be in a higher temperature (35-45 °C) and a narrower pH (2-3.5) range in the catalytic degradation of SMX (*Wan and Wang, J Hazard Mater, 2017,*

324, 653-664; Wan ang Wang, *J Chem Technol Biot.*, 2017, 92, 874-883. Bai et al., 2017, *Environ Prog Sustain.* 36, 1743-1753). However, we found that the catalytic degradation efficiency of SMT (5g/L) by sch-FLG was around 100% at a lower temperature (28°C), and in a wide range of initial solution pH value (1-9) in our study. It can be seen that sch-FLG has excellent catalytic performance and adapts to a wider pH range.

Table 1. Performance of the catalysts in other studies

Materials	Target pollutant	C ₀ mg L ⁻¹	H ₂ O ₂ mg L ⁻¹	catalyst g L ⁻¹	pH	T °C	time min	removal efficiency (%)	references
Fe ₃ O ₄ /Mn ₃ O ₄		20	204	0.5	2.5-3	45	50	100	[69]
Fe ₃ O ₄ /Mn ₃ O ₄ /rGO		20	285	0.5	3-3.5	35	80	98	[65]
Fe ₃ O ₄ magnetic nanoparticles	Sulfamethazine	20	680	1	2-3	-	150	80	[70]
sch-FLG		5	200	1	1-9	28	90	100	This work

Besides the above-mentioned contents, we tried our best to improve the manuscript and made some changes in the manuscript. These changes will not influence the content and framework of the paper and here we did not list the changes.

Appendix C

Manuscript number: RSOS-191977.R1

Title: Synthesis and assessment of schwertmannite/few-layer graphene composite for the degradation of sulfamethazine in heterogeneous Fenton-like reaction

Thanks the editor and reviewers for your critical review of our manuscript. The itemized responds to the Editor and reviewer's comments are as follows:

Comment 1: In reference 2, the use of accents and umlauts is not trivial in the names of the authors.

Reply: Thank you very much for your comment. We have checked this reference and corrected the miswriting of the author's name.

Comment 2: In reference 16, the use of hyphen is not trivial in the names of the authors.

Reply: Thank you very much for your comment. We have checked this reference and used the hyphen in the names of the authors.

Comment 3: In reference 26, the last name must be Diniz da Costa JC.

Reply: We are very sorry, and thank you very much for your comment. We have revised the last author name of the reference.

Comment 4: In reference 37, the use of accents must be using microsoft word not

using images or other font type.

Reply: We have checked and revised the names of the reference authors using microsoft word.

Comment 5: In references 69 and 70, the authors change the format of separating a name in two parts that they had used previously.

Reply: Thank you very much for your comment. We have checked and preserved the format of splitting names into two parts, which we have used before.

All of these are responses to comments, and our manuscript has been revised accordingly. Thank you very much again for your help in improving our manuscript's quality.